REGISTERED REPORT PROTOCOL

# Efficacy and safety of mirror therapy for post-stroke aphasia: A systematic review and meta-analysis protocol

Yufeng Peng[1], Shouqiang Huang[2], Xiaotong Yang[2], Jiao Ma[2]*

1 School of Medical Technology and Information Engineering, Zhejiang Chinese Medical University, Hangzhou, China, 2 Department of Emergency Medicine, Taihe Hospital, Hubei, China

* 18086429646@163.com

This is a Registered Report and may have an associated publication; please check the article page on the journal site for any related articles.

## Abstract

### Background

Aphasia is one of the most common complications of stroke. Mirror therapy (MT) is promising rehabilitation measure for the treatment of post-stroke aphasia. Although some studies suggested that MT is effective and safe for aphasia, the effects and safety remain uncertain due to lacking strong evidence, such as the relevant systematic review and meta- analysis.

### Methods

This study will search PubMed, Web of Science, Cochrane Library, EMBASE, Medline, China Knowledge Network (CNKI), WANFANG, China Biomedical Literature Database (CBM), from inception to 1[th] May 2023 to identify any eligible study. No language or date of publication shall be limited. We will only include randomised controlled trials of MT in the Treatment of poststroke aphasia. Two investigators will work separately on the study selection, data extraction, and study quality assessment. The western aphasia battery (WAB) and aphasia quotient (AQ) will be included as the main outcomes. Boston diagnostic aphasia examination method (BDAE), Chinese standard aphasia examination (CRRCAE) will be included as the secondary outcomes. The statistical analysis will be conducted by RevMan V.5.4 software. The risk of bias of included studies will be assessed by the Cochrane 'Risk of bias' tool. The quality of proof of the results will be evaluated by using the Grading of Recommendations Assessment, Development and Evaluation guidelines.

### Results

The finding will be presented in a journal or related conferences.

### Conclusion

This study will provide a basis for whether mirror therapy (MT) is effective and safe in the treatment of post-stroke aphasia.

**Data Availability Statement:** All relevant data are within the paper and its Supporting information files.

**Funding:** The authors received no specific funding for this work.

**Competing interests:** The authors have declared that no competing interests exist.

## Trial registration

### Systematic review registration

INPLASY registration number: INPLASY 202340054.

## Introduction

There are more than 10 million new cases of stroke worldwide each year, and at least one third of these patients present with aphasia [1,2]. Post-stroke aphasia is an acquired language disorder in which patients have impairments in all aspects of the language system [3], mainly including difficulties in auditory comprehension, spontaneous speech, repetition and naming [4]. Different patients have different manifestations of aphasia. Even within the same patient, symptoms related to aphasia can vary considerably, especially in the first weeks and months after stroke. The different types and severity of aphasia depend on many factors, including the size and location of the stroke, health status, time to recovery from stroke, and time since stroke [5]. Although many stroke survivors show spontaneous partial speech recovery, between one-third and one-half of these patients continue to suffer from speech dysfunction 6 months after the stroke [6]. Post-stroke aphasia can adversely affect daily life and work as well as communication skills and lead to a higher incidence of depression in patients [7]. However, treatment options for patients with this disorder remain relatively limited, such as speech therapy and pharmacological interventions. Therefore, it is crucial to develop new and effective approaches to treat this language disorder.

Mirror therapy is a kind of motor representation training based on action observation, visual imagery and imitation learning [8]. During the training process, a mirror device is used to copy the image of the activity of the healthy side of the limb to the affected side, and the patient stimulates the main motor cortex of the human brain through such visual feedback, which affects the electrical activity and excitability of the cortex and promotes brain function remodeling and induces motor function recovery [9]. Mirror neurons are specialized neurons with mapping functions, visual connectivity, and motor feedback; mirror neurons are activated both when performing motor tasks and when observing the same movements; therefore, mirror neurons are considered to be an important neural basis for understanding movements, imitating behaviors, language learning, and attention shifting [10,11]. In mirror training, by activating relevant mirror neurons, especially in the inferior frontal gyrus and pre motor cortex and inferior parietal lobule [12], thereby improving aphasia and spatial attentional function in stroke patients [13]. Many recent studies have attempted to use the mirror neuron theory to treat different forms of aphasia [13,14]. In one study, real MT was found to significantly improve language function in stroke patients compared to sham MT [15].

Mirror therapy has the advantages of safety, effectiveness, convenience and non-invasive, which can intervene in the early stage of stroke Aphasia and promote the recovery of speech function. Although numerous clinical studies have reported positive effects of mirror therapy on aphasia after stroke, there is no meta-analysis available. Therefore, this review conducts a meta-analysis to assess the efficacy and safety of mirror therapy for poststroke aphasia and provides a stronger basis for clinical treatment.

## Methods

This study will complete in accordance with the Preferred Reporting Initiative for Systematic Reviews and Meta-Analysis Project (PRISMA-P) guidelines [16]. This systematic review

program was registered on the INPLASY website. Registration number: INPLASY202340054。 As this study will be based on published studies, ethical approval is not required. The PRISMA-P checklist is attached as S1 File.

## Selection criteria

**Types of participants.** Patients with aphasia after stroke and older than 18 years old. All patients should be diagnosed with stroke by CT or MRI and confirmed by the clinician of aphasia presentation. However, gender, race and educational status are not restricted, to the exclusion of participants who are unable to cooperate with rehabilitation, such as hearing, visual and cognitive impairments or severe infections, organ malfunctions, etc.

**Types of interventions.** The interventions considered in the experimental group consist of mirror therapy. The control group will receive conventional speech rehabilitation therapy

## Types of outcomes

(1) The primary outcomes

The western aphasia battery (WAB) and aphasia quotient (AQ) will be included as the main outcomes. The WAB is an internationally standardized test of aphasia, and consists of four subtests: spontaneous speech, auditory comprehension, repetition, and naming. AQ will be calculated based on the aphasia quotient = (spontaneous speech + listening comprehension/20 + repetition/10 + naming/10) × 2.

(2) The secondary outcomes

Additional outcomes will be assessed using the Boston Diagnostic Aphasia Examination (BDAE), Chinese Standardized Review of Aphasia (CRRCAE), Communication Activities of Daily Living (CADL), Functional Assessment of Communication Skills (FACS) and Hamilton Depression Scale (HAM-D).

**Types of study.** This study will only include randomised controlled clinical trials (RCTs) of MT for poststroke aphasia. Studies will be excluded if they are conference papers, editorials, abstract opinions, case reports, and crossover studies.

## Search strategy

**Electronic searches.** We will search the following electronic databases from creation to 1 May 2023: Cochrane Library, EMBASE, Medline, Web of Science, Pubmed, the Chinese Biomedical Literature Database, the electronic databases of the WanFang Data and China National Knowledge Infrastructure. The following medical keywords: "mirror therapy", "aphasia", and " stroke" will be used in this search. No language or date of publication shall be limited. Details of the research strategy in the Pubmed database are presented in S1 Table.

**Searching other resources.** Other ongoing and unpublished clinical trial registry studies will be reviewed to find additional eligible studies.

## Data extraction and export

We will confirm the standard data extraction form prior to data extraction. Based on the requirements, the two researchers will independently review the literature. By reviewing the titles and abstracts, they will first use EndNote X 9.0 to weed out duplicate articles and remove those that don't meet the inclusion requirements. Secondly, they will read the full text based on the inclusion and exclusion criteria and screen the remaining articles again. Finally, they will register the excluded articles and explain the reasons and decide whether they are ready

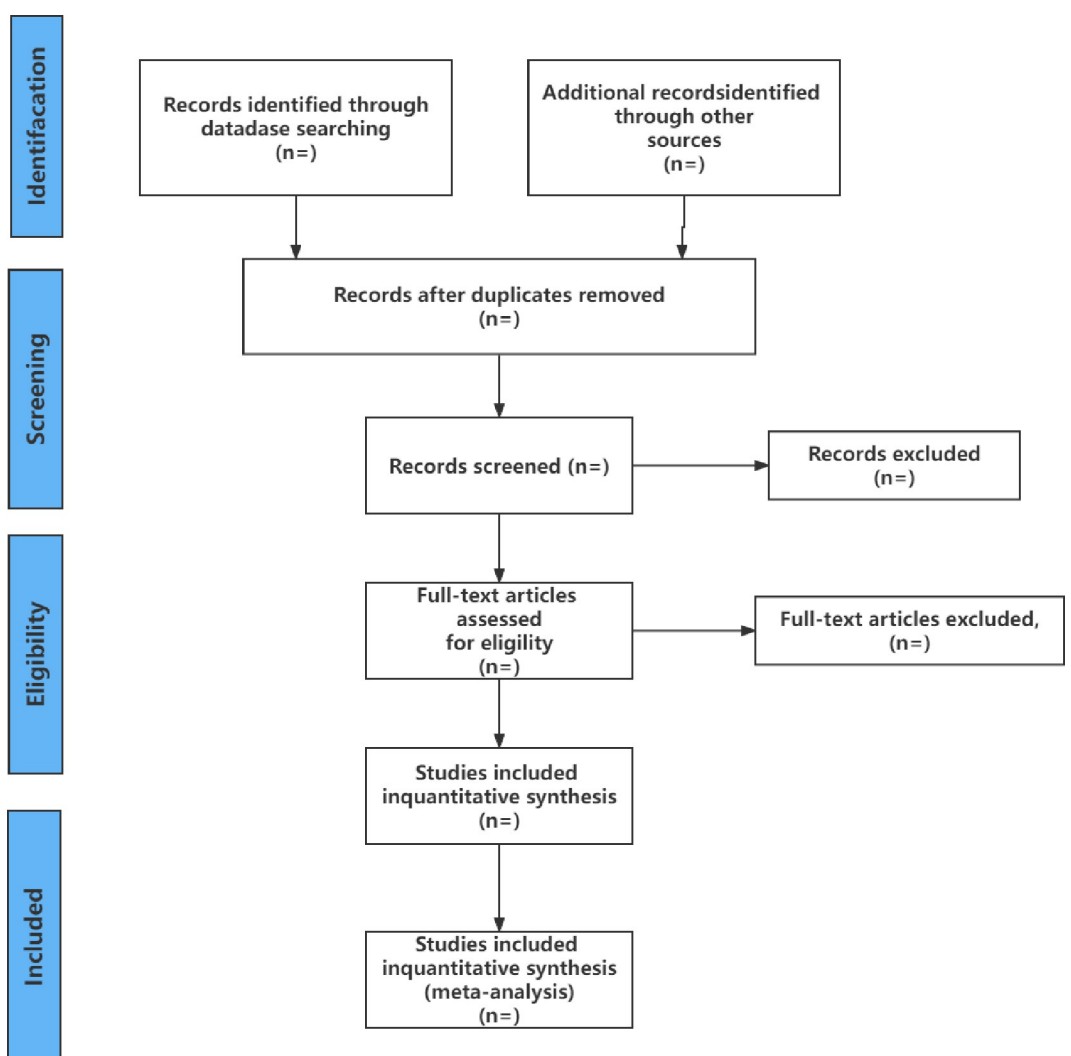

**Fig 1. Flow diagram of the study selection process.**

for a systematic analysis. In the event of a dispute, a third researcher will be asked for guidance and to mediate a resolution. Fig 1 depicts the exact screening procedure.

## Data extraction and analysis

The data will be extracted independently by two researchers using a predetermined standard Excel spreadsheet. This includes general information (authors, age, sex, country, race, year of publication, diagnostic criteria); study design (sample size, randomization and blinding details); details of interventions (treatment method, frequency, treatment time); and outcome indicators (primary and secondary outcomes, adverse events and others). Any disagreements between two investigators will be resolved by negotiation with a third investigator.

## Assessment of risk of bias in included studies

The quality of the included trials will be assessed using the Cochrane Collaboration's tool by two reviewers (PYF and HSQ). We will assess the risk of bias from the following seven

domains: 1. random sequence generation; 2. allocation sequence concealment; 3. blinding of participants and staff; 4. blinding of outcome assessment; 5. incomplete outcome data; 6. selective reporting; 7. other biases. Assessments will be categorized as low risk, high risk, and uncertain risk, and any disagreements between two investigators will be resolved by negotiation between a third investigator.

## Measures of treatment effect

If continuous data are included, mean difference (MD) or standard MD (SMD) will be used to calculate the treatment effect with a 95% confidence interval. If dichotomous data are present, the risk ratio (RR) and 95% confidence interval (CI) will be produced.

## Dealing with missing data

In order to obtain the missing data, we will make an effort to contact with the authors by phone or email. If still not available, we will perform an intention-to-treat analysis for missing participant data. At the same times, we will evaluate the potential impact for the missing data by performing sensitivity analysis.

## Assessment of heterogeneity

We will test heterogeneity by $I^2$ test. If $I^2$ is less than 50%, it means that there is no significant heterogeneity; if $I^2$ is greater than 50%, there is substantial heterogeneity. To investigate the causes of the heterogeneity in the study results, subgroup analysis and sensitivity analysis will be used.

## Data synthesis

The software Review Manager V.5.4 will be used to carry out the meta-analysis. Continuous data will be presented as the standardized mean difference (SMD) with a 95% confidence interval, whereas dichotomous data will be reported as the risk ratio (RR) with a 95% confidence interval (CI). The significance level will be set at 50% for the Higgins $I^2$ test, which will be used to examine heterogeneity. For the goal of meta-analysis, a model with fixed effects will be employed if heterogeneity is not considerable ($I^2 \leq 50\%$). If heterogeneity is significant ($I^2 \geq 50\%$), a random effects model will be used. We will perform sensitivity analysis and subgroup analysis to find their potential explanations.

## Assessment of publication biases

When more than ten studies are included, funnel plots will be generated to analyzed the potential reporting biases. If the distribution of funnel plot data is relatively symmetrical, that means no publication bias.

## Subgroup analysis

If possible, a subgroup analysis will be performed by the characteristics of the studies. Variations in the factors including intervention type, treatment duration, age, sample size and quality of studies, will be considered.

## Sensitivity analysis

To check the stability and dependability of the study's findings, the researchers will repeat the sensitivity analysis. One study at a time will be removed to confirm that the results are not

influenced by any one study. The outcomes of the data analysis will then be compared once more; if there is no change, the results are steady and trustworthy.

## Evidence quality evaluation

The quality of evidence for each outcome will be assessed using the Grading of Recommendations Assessment, Development and Evaluation guidelines [17]. The evaluation items mainly stem from following factors: the risk of bias, imprecision, inconsistency, indirectness, publication bias [17,18]. According to above factors, the assessments will be graded into very low, low, moderate or high level.

## Ethics and communication

Ethical approval is not required as individual patient data is not required in this study. The findings of this review will be presented at conferences or in peer-reviewed journal articles.

## Discussion

Aphasia is common following stroke, with frequencies ranging from 15 to 42 percent in acute settings and from 25 to 50 percent in chronic settings [19], and 32%-50% of patients with aphasia still have language impairment 6 months after stroke or even for the rest of their lives [20], which affects patients' communication, daily life, work and recreation to varying degrees. The efficacy of current medication or speech rehabilitation is limited, so it is important to explore effective treatment methods for post-stroke aphasia.

In recent years, the field of brain and cognitive science has developed rapidly, and mirror neurons are one of the most significant discoveries in this field [21]. Many new rehabilitation therapies based on mirror neuron theory, such as motor observation therapy, motor imagery therapy, mirror therapy, virtual reality therapy, and brain-computer interface technology, have been translated into stroke rehabilitation [22].

MT is based on the mirror neuron system, which uses the planar mirror imaging principle of contralateral active image projection onto the affected side. It also combines visual illusion, visual feedback and virtual reality, and also has the advantages of simplicity, convenience, effectiveness and no pharmacological side effects [23].

It has been shown to be effective in improving upper limb motor function in stroke patients with hemiplegia and reducing pain in patients with regional pain syndrome [23]. Furthermore, it has been shown that supplementing conventional speech rehabilitation with mirror therapy can improve the aphasia quotient of the Western Aphasia Battery in patients with aphasia [24]. In addition, MT training in the acute post-stroke period can facilitate the recovery of language function in patients with motor aphasia. This intervention significantly strengthened the functional connectivity between the frontal, temporal, and parietal lobes of the left hemisphere brain and reinforced the connection with the hippocampus, suggesting that MT training can promote language recovery and reveal the underlying neuroplasticity mechanism of language recovery after stroke [15].

However, there is no systematic review and meta-analysis on this issue, The effectiveness and safety of MT in adult patients with post-stroke aphasia will thus be the focus of the first systematic review and meta-analysis in this field. This study offers scientifically sound medical support for this disease's treatment. The study has a number of restrictions. First, heterogeneity may exist because most of the included studies differed in method selection, site selection, treatment frequency, and treatment duration. Second, most of the literature included in this study was in Chinese, which may have led to language selection bias.

## Supporting information

**S1 Table. Search strategy.**
(DOC)

**S1 File. PRISMA-P-checklist.**
(DOC)

## Author Contributions

**Resources:** Xiaotong Yang.

**Software:** Shouqiang Huang.

**Writing – original draft:** Yufeng Peng.

**Writing – review & editing:** Jiao Ma.

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
