## [Decision Letter · Decision Letter 0]

11 Jan 2024

PONE-D-23-24347Efficacy and Safety of Mirror Therapy for Post-stroke Aphasia: A Systematic Review and Meta-analysis ProtocolPLOS ONE

Dear Dr. Jiao Ma,

Thank you for submitting your manuscript to PLOS ONE. After careful consideration, we feel that it has merit but does not fully meet PLOS ONE’s publication criteria as it currently stands. Therefore, we invite you to submit a revised version of the manuscript that addresses the points raised during the review process.

We look forward to receiving your revised manuscript.

Kind regards,

Nadinne Alexandra Roman, Ph.D.

Academic Editor

PLOS ONE

Journal Requirements:

2. In your cover letter, please confirm that the research you have described in your manuscript, including participant recruitment, data collection, modification, or processing, has not started and will not start until after your paper has been accepted to the journal (assuming data need to be collected or participants recruited specifically for your study). In order to proceed with your submission, you must provide confirmation.

3. Please include a copy of Table 1 which you refer to in your text on page 4. 

Reviewers' comments:

Reviewer's Responses to Questions

**Comments to the Author**

1. Does the manuscript provide a valid rationale for the proposed study, with clearly identified and justified research questions?

Reviewer #1: Yes

Reviewer #2: Yes

2. Is the protocol technically sound and planned in a manner that will lead to a meaningful outcome and allow testing the stated hypotheses?

Reviewer #1: Yes

Reviewer #2: Yes

3. Is the methodology feasible and described in sufficient detail to allow the work to be replicable?

Reviewer #1: Yes

Reviewer #2: Yes

4. Have the authors described where all data underlying the findings will be made available when the study is complete?

Reviewer #1: Yes

Reviewer #2: Yes

5. Is the manuscript presented in an intelligible fashion and written in standard English?

Reviewer #1: Yes

Reviewer #2: Yes

6. Review Comments to the Author

You may also provide optional suggestions and comments to authors that they might find helpful in planning their study.

Reviewer #1: I appreciate the opportunity to review this protocol. This is an interesting topic; however, the introduction needs to be revised to improve clarity.

1. Please provide a correct definition of aphasia. Aphasia is a language disorder and speech can be affected in aphasia as aphasia can co-occur with apraxia and dysarthria.

"Aphasia is characterized by slowed speech, distorted phonemes, and a tendency for speech to separate into individual syllables with balanced stress between adjacent syllables [4]". This is an incorrect definition of aphasia. These are features of apraxia and dysarthria.

Aphasia is a language disorder and there are different types of aphasia. For this protocol, it appears that the authors are discussing about non-fluent aphasia.

2. Please provide a clear rationale as to why mirror therapy may benefit patients with aphasia. Mirror therapy was originally developed for improving motor function.

3. The primary and secondary outcomes as assessing aphasia as a global measure ( e.g., WAB, BDAE etc). It is also important to determine whether mirror therapy has an effect on quality of life or functional communication.

Reviewer #2: Dear Authors:

Thank you for allowing us to read your interesting manuscript about a systematic review and meta-analysis protocol to investigate the efficacy and safety of mirror therapy for post-stroke aphasia. Please consider the following comments:

1.- In the section “2.1.1 Types of participants” the age group of the patients to be included is not mentioned. Throughout the text, age of patients is only mentioned in the following paragraph of the discussion “The effectiveness and safety of MT in adult patients with post-stroke aphasia will thus be the focus of the first systematic review and meta-analysis in this field”. I suggest clarifying this point.

2.- In the first paragraph of the discussion, you mention the number of patients with aphasia secondary to stroke and traumatic brain injury, however, patients with trauma will not be included in this study, so I suggest that the text refers only to stroke patients.

Thanks

7. PLOS authors have the option to publish the peer review history of their article (what does this mean?). If published, this will include your full peer review and any attached files.

Reviewer #1: No

Reviewer #2: **Yes: **Vanessa Cano-Nigenda

---

## [Author Response · Author response to Decision Letter 0]

16 Feb 2024

Dear Dr. Nadinne Alexandra Roman,

I have uploaded Cover Letter and A Rebuttal Letter.

---

## [Decision Letter · Decision Letter 1]

17 Mar 2024

Efficacy and safety of mirror therapy for post-stroke aphasia: A systematic review and meta-analysis protocol

PONE-D-23-24347R1

Dear Dr. Jiao Ma,

We’re pleased to inform you that your manuscript has been judged scientifically suitable for publication and will be formally accepted for publication once it meets all outstanding technical requirements.

Kind regards,

Nadinne Alexandra Roman, Ph.D.

Academic Editor

PLOS ONE

Additional Editor Comments (optional):

Reviewers' comments:

Reviewer's Responses to Questions

**Comments to the Author**

1. Does the manuscript provide a valid rationale for the proposed study, with clearly identified and justified research questions?

Reviewer #1: Yes

Reviewer #2: Yes

2. Is the protocol technically sound and planned in a manner that will lead to a meaningful outcome and allow testing the stated hypotheses?

Reviewer #1: Yes

Reviewer #2: Yes

3. Is the methodology feasible and described in sufficient detail to allow the work to be replicable?

Reviewer #1: Yes

Reviewer #2: Yes

4. Have the authors described where all data underlying the findings will be made available when the study is complete?

Reviewer #1: Yes

Reviewer #2: Yes

5. Is the manuscript presented in an intelligible fashion and written in standard English?

Reviewer #1: Yes

Reviewer #2: Yes

6. Review Comments to the Author

You may also provide optional suggestions and comments to authors that they might find helpful in planning their study.

Reviewer #1: The authors have addressed the concerns raised by the reviewer in the revised version of the manuscript.

Reviewer #2: Dear authors

We appreciate that you took our comments into account and made the suggested adjustments. No more comments

7. PLOS authors have the option to publish the peer review history of their article (what does this mean?). If published, this will include your full peer review and any attached files.

Reviewer #1: No

Reviewer #2: **Yes: **Vanessa Cano-Nigenda
